# Consumer Preference and Purchase Intention for Plant Milk: A Survey of Chinese Market

**DOI:** 10.3390/foods14071240

**Published:** 2025-04-01

**Authors:** Aili Wang, Chunhua Tan, Wenwen Yu, Liang Zou, Dingtao Wu, Xuanbo Liu

**Affiliations:** 1Key Laboratory of Coarse Cereal Processing (Ministry of Agriculture and Rural Affairs), College of Food and Biological Engineering, Chengdu University, Chengdu 610106, China; tanchunhua@cdu.edu.cn (C.T.); yuwenwen@cdu.edu.cn (W.Y.); zouliang@cdu.edu.cn (L.Z.); wudingtao@cdu.edu.cn (D.W.); 2Department of Food Science and Technology, Virginia Polytechnic Institute and State University, 1230 Washington Street SW, Blacksburg, VA 24061, USA; xuanbol@vt.edu

**Keywords:** plant milk, consumer, behavior, purchase intention, Chinese market, nutrition

## Abstract

Plant milks are considered to be nutritious, sustainable, and vegetarian food products, and they have been the fastest growing beverages in the past decade in China. However, few studies have investigated consumers’ demands and purchase behaviors with respect to plant milks. Through an online questionnaire (n = 1052 valid responses), this study identified the factors that influenced individuals’ purchase intentions, purchase behaviors, attitudes, and demands with respect to current and future plant milk products. Through descriptive analysis and PCA, this study revealed that nutritional value (63.6%), taste (56.3%), and calories (42.8%) were the top three factors that Chinese consumers most cared about regarding plant milks. In the current Chinese market, coconut milk is the most popular plant milk with the highest purchase rate (61.2%), followed by soymilk (53.9%). Male consumers preferred plant milk with higher protein content and fortified with antioxidants, while female consumers preferred plant milk low in calories and enriched with collagen, dietary fiber, and probiotics. Chinese consumers are willing to pay higher prices for plant milks with enhanced nutritional value, improved product quality, and strengthened safety assurances. Innovative forms of plant milk, such as bean milk, rice milk, and quinoa milk, may be developed to satisfy the diversified needs of consumers.

## 1. Introduction

Plant milks can be made from various plant sources, such as soy, almonds, coconut, or oats, which are considered to be alternatives to traditional dairy milk. Plant milks are good sources of macronutrients and micronutrients, including protein, amino acids, vitamins, unsaturated fatty acids, and other nutrients [1,2]. Additionally, plant-based beverages such as soymilk are low in saturated fat and do not contain cholesterol. Thus, plant milks are often presented as healthy, sustainable, and vegetarian food products, and they are commonly classified into four categories: cereal-based, legume-based, nut-based, and pseudocereal-based plant milks [3,4,5].

The main driving forces of plant milk consumption stem from physiological conditions (lactose intolerance, dairy milk allergy, cholesterol), lifestyle choices (vegan diet, high-protein diet), and food safety issues (growth hormone or antibiotic residues in cows’ milk) [6,7]. In China, consumers experienced the melamine incident (milk scandal) in 2008, which undermined public confidence in traditional dairy milk. Therefore, plant milk has begun to be favored by consumers and has been the fastest growing beverage, with a ten-year average growth rate of 24.5% since 2008 [8]. The production of plant milk was 8.7 million tons in 2019, and the plant milk market is expected to exceed USD 300 billion in 2025, showing a positive growth trend [8,9].

A group of consumers willing to pay a premium for higher-quality and more nutritious food has emerged in China [9,10]. However, few studies have investigated consumers’ demands and expectations of plant milk products. Furthermore, the current plant-based beverages in the Chinese market are highly homogeneous, lacking the innovation and diversification to satisfy the diversified needs of consumers. Few studies have analyzed consumers’ demands with respect to the future development of plant milk. Therefore, this study aims to identify the factors that influence individuals’ purchase intentions and purchase behaviors with respect to plant milk, to understand consumers’ attitudes to current plant milk products, and to demonstrate consumers’ demands with respect to future plant milk products. Through an online questionnaire survey and data analysis, the findings of this study are expected to reveal consumers’ motivations for the purchase and consumption of plant milk, and to provide suggestions for the plant milk industry to improve the current products. This study also attempts to shed new light on the future development of plant-based beverages in China.

## 2. Materials and Methods

### 2.1. Participants

Ethical approval for the involvement of human subjects in this study was granted by the Chengdu University Research Ethics Committee (#21020231009) in October 2023. This study utilized random sampling to collect responses through the distribution of an online survey, and informed consent was obtained from each subject prior to their participation in the study. For minors (participants under 18 years of age), the informed consent was obtained from their parents. The survey was designed and administered using the Wenjuanxing software accessed at https://www.wjx.cn/app/themehtml/wjxai.aspx accessed on 23 January 2024, (Ranxing Information Technology Co., Changsha, China) to collect data. The survey was accessible from 1 January 2024 to 15 December 2024. There were a total of 1153 respondents who participated in answering the questionnaires about Chinese consumers’ perceptions of plant milk. Among them, 1052 valid questionnaires (91%) were successfully completed. The participants provided informed consent via the statement “I am aware that my responses are confidential, and I agree to participate in this survey”, where an affirmative reply was required to enter the survey. They were able to withdraw from the survey at any time without giving a reason. This study was explained to the consumers in the online questionnaire. All participants were informed that they would participate in the survey using their personal smartphone or notebook, and that all data would be de-identified and only reported in the aggregate. The demographic information of the respondents in this study, including gender, age, education, income, occupation, and location, were recorded.

### 2.2. Questionnaires

The questionnaire used in this study was designed in four parts that related to consumers’ perceptions of plant milk beverages, including cognitive analysis, purchase intention, purchase behavior, and user needs and product demands of plant milk. The design of the questionnaire was based on previous publications, with modifications [11,12,13]. Before answering the questions, participants were provided with a description of plant milk.

In the cognitive analysis section of this study, the respondents were asked to rate their familiarity with plant milk (not familiar at all, somewhat familiar, familiar, very familiar). Besides, respondents were asked “Why do you like plant milk? (good taste, high protein, low calories, vegetarianism, low price, nutritional value)”, “What are the defects of plant milk (bland taste, insufficient protein content, limited nutrients, too sweet, fewer flavor)”, and “What brands of plant milk do you know [Six walnuts (walnut milk), Yinlu (peanut milk), Yeshu (coconut milk), Vita-milk (soymilk), Lulu (almond milk), Oatly (oat milk), other]?”.

To analyze purchase intention for plant milk, respondents were questioned about “What types of beverages do you usually drink (bottled water, fruit juice, protein beverages such as plant milk, tea beverages, carbonated beverages such as soda, vegetable juice, dairy beverages, fermented beverages such as yogurt or probiotic drink)?”, “What are the decisive factors for your beverage selection (taste, price, nutritional value, calories, protein content) ?”, “How often do you drink plant milk (occasionally, every day, every week, every month)?”, “What is your perception about the price of current plant milk?” using a 5-point scale ranging from “very cheap” (1) to “very expensive” (5), and “How much are you willing to pay for the plant milk (250 mL) (1–3 yuan, 4–6 yuan, above 7 yuan, does not matter) ?”.

To analyze consumers’ purchase behaviorsof plant milk were inquired by questions including “Why do you purchase plant milk? (good taste, quenches thirst, attractive packaging, nutritious, on-sale, celebrity endorsement, other)”, “What types of plant milk do you usually buy? (walnut milk, peanut milk, coconut milk, soymilk, almond milk, oat milk, other)”, “Where do you purchase plant milk? (e-commerce platforms, supermarkets/grocery stores, convenience stores, restaurants)”, and “What are the decisive factors for your plant milk selection? (taste, price, nutritional value, brand)”.

In order to understand consumers’ demand for plant milk and predict the future consumption trends, the following questions were set in the survey including “What types of plant milk beverages would you like to see in the future? (rice milk, sweet potato milk, nuts milk such like pistachio milk, bean milk such like chickpea milk)”, “What features of plant milk would you like to see in the future? (higher protein content, lower calories, higher dietary fiber, vitamin enrichment, minerals enrichment, collagen enrichment, probiotic enrichment, antioxidant enrichment)”, and “For which factors would you be willing to pay a higher price for plant milk? (better taste, higher nutritional value, enhanced food quality, strengthened food safety, well-known brand)”.

### 2.3. Statistics

This study employed the quantitative research method, and the data were obtained from an online questionnaire. Statistical analysis methods included the descriptive statistics (frequency distribution tables and percentages) and principal component analysis (PCA) to analyze the relationships among various factors influencing consumption and purchase behaviors of plant milk. For questions allowed for multiple-choice responses, a binary system was used: “1” for selected options and “0” otherwise. Prior to performing PCA, a correlation matrix was calculated for categorical variables to ensure appropriate data preparation. For multiple-choice questions, the frequency of selection for each option was calculated by dividing the count of responses for that option by the total number of responses to the question.. All data analyses were performed using Microsoft Excel (2021) and JMP Pro (Ver 17.0.0, SAS Inc., Cary, NC, USA).

## 3. Results

### 3.1. Demographic Information

The demographic information of the participants in this study was summarized in Table 1. Among the 1052 respondents, there were 646 females (61.4%) and 406 males (38.6%). Most of the respondents were aged between 18–30 years of old (65.1%), while the remaining respondents were below 18 years of old (7.0%), 31–40 years of old (16.6%), 41–45 years of old (7.5%), and above 50 years of old (3.7%). About half of the participants were students (55.51%) with a monthly salary below 3000 yuan (58.9%). The majority of the participants held a bachelor’s degree (67.0%). The largest proportion of the participants were resided in southwest China, which was about 41.4%.

### 3.2. Cognitive Analysis of Consumers

Cognitive analysis allows researchers to explore consumers’ thought processes, perceptions, and preferences related to plant milk. By asking targeted questions, the survey identified the factors that significantly impact consumers’ willingness to purchase plant milk. The questionnaire began by assessing participants’s familiarity with plant milk. This baseline assessment excluded participants who were completely unfamiliar with plant milk, resulting in a valid sample size of 1,052 participants. Figure 1a showed the participants’ familiarity with plant milk. The majority (79.9%) of the respondents were familiar or very familiar with plant milk.

Reasons for liking and disliking plant milk were analyzed and presented in Figure 1b,c. Nutritional value was the most important attribute for consumers, followed by taste, calories, price, protein content, and vegetarianism. Although one of the key features of plant milk is high protein content, it is noteworthy that Chinese consumers prioritized nutritional value, taste, and calories over protein content. As shown in Figure 1c, the limited variety of flavors was identified as the primary defects of plant milk, followed by excessive sweetness, insufficient nutrients, and lower-than-expected protein content. According to the results, most of the consumers choosing plant milk were based on their demand for pursuing a healthy diet. However, sensory experience also significantly influenced consumers’ purchasing decisions.

Consumers’ awareness of different brands of plant milk in this study was summarized in Figure 1d. Six walnuts (walnut milk, Hengshui, Hebei, China) showed the highest visibility among the plant milk, followed by Yinlu (peanut milk, Xia’men, Fujian, China), Vitasoy (soymilk, Hongkong, China), Coconut Palm (coconut milk, Haikou, Hainan, China), Viee (walnut and peanut milk, Chengdu, Sichuan, China), Lolo (almond milk, Chengde, Hebei, China), and Oatly (oat milk, Ma’anshan, Anhui, China). At present, there isn’t a dominant brand effect in the Chinese plant-based protein beverage market, allowing multiple brands to coexist harmoniously.

### 3.3. Purchase Intention of Consumers

Analyzing the purchase intention of consumers is essential for identifying the drives and barriers to a product and predicting the future consumer behavior. The preferences of respondents when making a puchase were summarized in Figure 2a. Around half of the respondents selected fruit juice when choosing a drink. More than 40% of the respondents preferred tea beverages, followed by bottled water, plant milk, fermented beverages, and soda. Less than 20% of the participants chose dairy beverage and vegetable beverage. Figure 2b showed the decisive factors influencing beverage selection. Both taste and prices wereprioritized by more than 70% of participants, making them the most important factors affecting purchase intentions. Given a favorable taste and reasonable price, consumers preferred beverages with nutritional value and low calories. Less than 30% of participants emphasized protein content as a key factor in their beverage selection.More than half of the respondents (53.8%) considered the current price of plant milk to be acceptable (Figure 2c). However, 34.9% of the respondents believed the price of current plant milk was higher than expected. As shown in Figure 2d, more than half of the respondents indicated that a price range of 4–6 yuan was appropriate for plant milk beverages.

### 3.4. Purchase Behavior of Consumers

Analysis of consumers’ purchase behaviors provides valuable insights for product development, price strategies, and future market predictions. The consumption frequency of plant milk was shown in Figure 3a. About half of the respondents consumed plant milk monthly and 34.1% consumedplant milk weekly. Daily consumption of plant milk was 5.7% of the population. Purchase frequency of plant milk was relatively higher for coconut milk and soy milk (>50%), while lower for almond milk and oat milk (<20%) (Figure 3b). It is interesting to find that although Six Walnuts (walnut milk) and Yinlu (peanut milk) were the most well-known brands among respondents, the majority of the respondents purchased coconut and soy milk more often than the walnut and peanut milk.

When asked about the decisive factors for selecting plant milk, most of the participants selected the sensory perception (good taste) as their primary concern (Figure 3c), which might explain the high purchase frequency of coconut and soy milk. More than half of the participants considered nutritional value an important factor when choosing plant milk. About 30% of the participants purchased plant milk due to its low price (on-sale). Additionally, respondents were asked where they commonly purchased plant milk. As shown in Figure 3d, the majority of respondents purchased plant milk from supermarkets, grocery store, convenience stores, or e-commerce platforms, indicating that the plant milk might be part of their regular consumption. Only 21.5% of respondents chose plant milk when dining out, mainly paired with food.

### 3.5. Consumers’ Demands of Future Products

In order to understand the demands of consumer and predict the future development of plant milk, participants were inquired about their expectations for future plant milk products. As shown in Figure 4a, more than 60% of respondents expressed interest in the development of nut-based milk, such as cashews, pistachio, hazelnut, or pine nut milk. Additionally, more than half of the respondents expressed a desire for potato-based (sweet potato and purple potato) and rice-based (glutinous rice and purple rice) beverages.

Regarding the desired features of plant milk in the future, more than half of respondents anticipated the development of plant milk with lower calorie content and higher dietary fiber. This preference may stem from the increasing incidence of health concerns such as diabetes, obesity, hyperlipidemia, and constipation caused by excessive carbohydrate intake and insufficient dietary fiber in recent years in China. Besides, more than 40% of respondents expressed interest in plant milk products with higher protein content or enriched with vitamins. In contrast, addition of antioxidants, collagen, probiotics, and minerals was considered less important in plant milk (Figure 4b).

As shown in Figure 4c, consumers showed a willingness to pay higher price for plant milk with enhanced nutritional value, improved product quality, enhanced food safety, and better taste. Only a small portion of respondents would like to pay more only for brand recognition.

## 4. Discussion

### 4.1. Consumers’ Awareness and Attitude of Plant Milk

Although Chinese consumers demand product features similar to those of Western consumers, such as brand, quality, and flavor, Chinese consumers strongly emphasize on the nutritional value of food products [14]. Lee et al. [15] reported that about 65% of Chinese consumers believed the nutritional value of food products was positively associated with health and safety, and favored that healthy drinks rich in nutrients such as calcium and vitamin [15]. In our study, respondents’ choice of plant milk was driven by the same reason. As shown in Figure 1b, over 60% of consumers preferred plant milk due to its high nutritional value. Furthermore, Giacalone et al. [16] highlighted that sensory attributes significantly influence the acceptance of plant-based beverages. Our study supports this observation, with taste (56.3%) and low calories (42.8%) being the top factors valued by consumers. Su et al. [8] also found that nutritional value is a significant driver affecting consumer’s choice when selecting plant milk, with preference given to products rich in protein, vitamins, and minerals. Furthermore, health benefits, such as being lactose-free and low in saturated fat and cholesterol, also play a crucial role [8]. However, our results indicate that Chinese consumers may prioritize general health benefits over specific nutrient contents.

We summarized the nutrition facts label of major brands of plant milk in both Chinese and American market. As shown in Table 2, the primary nutritional value of Chinese plant milk was originated from its protein content; only two types of plant milk are enriched with vitamins. Soymilk (Vitasoy) contains VB2 (0.11 mg/100 mL), VB6 (0.11 mg/100 mL), and VB3 (1.05 mg/100 mL). Peanut milk (Viee) contains VE (0.7 mg/100 mL). Although carbohydrates and fats provide energy, consumers generally believe that reducing the intake of these components is essential for managing weight and lowering the risk of heart disease [17]. However, only 33.3% of respondents in our study chose plant milk for its high protein content (Figure 1b). One possible explanation might be the consumers knew the nutritional composition of the raw ingredients (e.g., grains or nuts), even though these nutrients were not labelled in the product. For example, almonds are known to contain various nutrients, such as protein (~20 g/100 g), vitamin A, vitamin E, calcium and iron [18]. Thus, consumers might select certain raw materials rather than the processed plant milk. Another explanation might be that consumers were unable to identify the specific nutrients they desired. Instead, some consumers might be chasing for a general “healthy food” rather than certain nutrients.

Among the raw materials of plant milk listed in Table 2, soybean contained the highest content of protein (~36 g/100 g), followed by peanut (~26 g/100 g), almond (~20 g/100 g), oat seed (~15 g/100 g), walnut (~15 g/100 g), and coconut (~3 g/100 g) [18,19,20,21,22,23]. However, the protein content of plant milk shown in Table 2 did not follow this order. The protein content in oat milk was higher than that in almond milk. Furthermore, the protein content in coconut milk was much higher than expected, which was close to the walnut milk. This variation might arise from the different ratios of raw materials to water across different products.

In addition to nutritional value, taste (56.3%) and low calorie (42.8%) of plant milk were the most important factors influencing consumers preferences for plant milk. In contrast, the top three defects of current plant milk products were bland flavor (45.2%), too sweet (44.7%), and limited nutrients (43.0%). Therefore, nutritional value, taste, and calories were the top three concerns for Chinese consumers regarding plant milk. 

### 4.2. Consumer Behavior and Purchase Intentions of Plant Milk

The top two influencing factors for selecting a beverage were taste (77.8%) and price (73.5%) in our study, which was consistent with previous publications on consumer preferences [3,8]. Su et al. [8] reported that sensory attributes such as taste, texture, and overall sensory experience are critical for consumer acceptance, as products with pleasant taste and texture are more likely to be preferred. Furthermore, price remains a key factor in consumer decisions, with some consumers willing to pay a premium for high-quality products, while others areprice-sensitive [8]. Although the current prices of plant milk are generally acceptable, there is a growing demand for higher quality and innovative products at reasonable prices [10]. However, when respondents were asked about the decisive factors influencing their plant milk purchases, taste (60.8%) and nutritional value (52.0%) emerged as the most important considerations. This result suggested that Chinese consumers might consider plant milk to be nutritious beverage and their choice may stem from a pursuit of nutritional value without being overly affected by the price.

More than half of the respondents felt the current prices of plant milk were acceptable and a price range of 4–6 yuan/250 mL of plant milk was reasonable (Figure 2c,d). We compared the market prices of plant milk form one of China’s largest e-commerce platforms (Taobao) and Walmart in USA. As shown in Table 2, most plant milk products in China fell within the price range of 2–4 yuan, aligning with consumers’ expectations. Among them, oat milk had the highest unit price while the soymilk held the lowest unit price. Oat milk (Oatly) entered the Chinese market in 2018 through partnerships with upscale coffee shops,, which contributed to its rising popularity and higher price. In the US market, coconut milk had the highest unit price while almond milk had the lowest unit price (Walmart). After investigating the price of raw materials of plant milk, it is interesting to find that almond had the highest unit price while soybean had the lowest unit price both in China and the US. However, the retail price of almond milk and soymilk were not consistent with the price of raw materials. This discrepancy might be attributed to the different processing technologies, machines, and liquidity ratio, which determined the quality and price of final product.

As shown in our study, most respondents consumed plant milk a few times per month, while 34.1% of respondents drank plant milk weekly. It is important to note that the inquires in this study focused on the packaged plant milk and did not include homemade soy milk or soy milk from street stalls.. As the most popular breakfast beverage in China, soymilk is widely consumed throughout the nation, particularly freshly made soy milk that can be purchased individually. Therefore, plant milk shows significant market prospect in China. Furthermore, due to the long-standing dominance of soymilk as a traditional breakfast with limited variety, other types of plant milk show strong growth potential in China.

In addition, other factors also affected consumer preference and purchasing behaviors. For example, brand awareness and trust play a significant role in shaping preferences, with well-known and trusted brands more likely to be chosen [3,8]. In our study, although Six Walnuts showed the highest visibility among plant milk brands, there isn’t a dominant brand effect in the Chinese plant-based protein beverage market. This presents an opportunityfor various brands to capture market share by highlighting unique selling points and addressing specific consumer preferences. Furthermore, social influence and current trends in plant-based diets also shape consumer acceptance and preferences [8]. Social factors, such as recommendations from friends and family, as well as current trends in plant-based diets, can impact consumer acceptance and preference for plan milk. In addition, social influence, knowledge and advertising exposure positively affect the purchase frequency and the price willing to pay [9]. In the future study, the survey may include questions to capture the impact of social influence and trends, such as “Have you ever purchased plant-based milk based on a recommendation from someone you know?”, “Have you noticed an increase in discussions about plant-based diets in social media, news, or other media sources?”, “Do you feel that plant-based milk is becoming more popular in your community or social circles?”.

Among current plant milk products, coconut milk was the most popular choice among respondents, followed by soy milk. As shown in Figure 3c, taste was the most decisive factor influencing plant milk purchases. Coconut milk has a rich, creamy texture and an unique coconut flavor that appeals to consumers. Although soybean has a distinct bean flavor, soymilk is widely accepted in China as a traditional staple. Oat milk has a special wheat aroma with creamy texture, however, it entered the Chinese market as a coffee/milk tea mate rather than an individual beverage. Thus, the sale and consumption of oat milk still low in retail channels. Almond milk was once highly popular in China, however, the emergency of new varieties of plant milk especially walnut milk have replaced it.

### 4.3. Consumer Demands and Future Development for Plant Milk

Innovation in flavors and formulations has the potential to attract consumers, such as almond, soy, oat, and coconut milk, caters to different consumer preferences [5]. For the future development of new variety of plant milk, our survey provided several directions including nut milk, potato milk, rice milk, and bean milk. It is interesting to find that consumer preferences for these plant milk types were almost evenly distributed, with a slightly higher demand for nut-based beverages. In the U.S. market, cashew milk is already available in the market. Future nut milk varieties, such as pistachio milk, chickpea milk, and macadamia nuts milk, may also be developed as plant milk with distinct flavor and rich nutrients. Meanwhile, new types of bean milk (black bean milk and red bean milk) have already appeared in the Chinese market. Accompanied by the increase of obesity and diabetes in China, the public has realized the consequence of excessive intake of carbohydrates and fats and started to adjust their dietary structure by increasing the intake of protein and dietary fiber. Consequently, whole grains such as buckwheat, quinoa, and highland barley are promising raw materials for plant milk.

The increasing incidence of obesity and diabetes in China has led to a growing interest in plant milk with lower calories and higher dietary fiber. This trend is consistent with global shifts toward healthier diets [24]. Our study suggests that significant potential for the development of diverse plant milk varieties, such as nut milk, potato milk, rice milk, and bean milk, to meet diverse consumer demands. Additionally, gender-specific preferences observed in our study align with previous research indicating that female consumers prioritize beauty and skincare benefits, while male consumers focus on muscle building and fitness [25,26]. These insights provide valuable guidance for targeted product development and marketing strategies. By integrating these discussions with our findings, we provide a comprehensive analysis that not only reiterates our results but also places them within the broader context of existing literature.

As shown in Figure 5, PCA analysis was conducted to examine the relationship between plant milk characteristics and consumer demographics. Component 1 explained 62.1% of the variance, and component 2 explained 18.7%. Notably, consumers’ gender was positively associated with specific attributes of plant milk. We have observed that female consumers showed the greatest interest in plant milk with high dietary fiber, enriched collagen, and probiotics. Collagen supplements have gained significant popularity in the Asian market, particularly among females [25]. Collagen is a protein that plays a crucial role in maintaining skin firmness, elasticity, and hydration, which helps to decrease the wrinkles and sagging skin. The Asian female consumers who prioritize skincare chasing for collagen supplements appealing for maintaining youthful skin [25]. In addition, probiotics has been reported to benefit the balance of microflora in the digestive tracts, which prevent multiple diseases to the host. Consumption of commercial probiotics keep increasing in recent years [26]. 

In contrast, male consumers preferred food/beverages with high protein and enriched antioxidants, driven largely by their interest in muscle building and fitness [27]. They also considered “taste” to be an important factor in their food choices. Besides, we also identified a cluster of consumers who favored plant milk with low calories and attractive packaging. This group was primarily composed of vegetarians with higher levels of education (master’s or doctoral degrees). Several studies have demonstrated a positive correlation has been demonstrated between higher educational level and the choice of being vegetarian as well as adoption of low-calorie diet [24].

With rapid urbanization and a rising middle class, Chinese consumers are increasingly attentive to environmental issues and health concerns, such as animal welfare and environmental sustainability, making them receptive to ethically produced alternatives. In China, individuals with a strong environmental consciousness are more willing to purchase plant milk, especially urban consumers who are more influenced by sustainability, animal welfare, and health concerns compared to their rural counterparts [28]. The country’s vibrant digital ecosystem and the profound influence of social media further suggest that positive media messages and social endorsements can effectively boost product interest. Positive media messages about plant-based food influence consumers’ purchase intentions. Social influence exerts a considerable impact on consumer behavior, implying that marketing strategies could capitalize on social networks to enhance the appeal and drive the adoption of plant-based milk alternatives [5,9]. Grasping these factors is vital for shaping future research approaches that effectively enhance plant-based consumption.

Consumer curiosity and willingness to try new products are strong drivers in the plant milk market. Consumer acceptance varies by ingredient—for example, almond, oat, and pea-based alternatives are preferred over cashew-based ones, while coconut and soy perform comparably favorably [4]. Sensory attributes such as sweetness, creaminess, smoothness, and appearance can increase the purchase intention of consumers. One promising strategy emerging from the literatures is the blending of multiple plant sources—either in their natural form or as isolated protein fractions—to enhance the overall sensory profile without relying on added flavors or additives. However, tailoring product formulations to align with familiar tastes while emphasizing health-oriented attributes may be more feasible in increasing consumer acceptance, such as mixing coconut flavor with new plant milk, making these beverages not only more appealing but also a natural extension of consumers’ lifestyle choices in China.

## 5. Conclusions

In the current Chinese market, coconut milk is the most popular plant milk with the highest purchase rate, followed by soymilk. New varieties of plant milk may be developed to meet the diversified needs of consumers, such as nut milk (pistachio, hazelnut, or pine nut milk), potato milk (sweet or purple potato milk), rice milk (glutinous or purple rice milk), and bean milk (chickpea, red, or green bean milk). In addition, new flavors such fruit, chocolate, and coffee flavors can be added to plant milk.

Chinese consumers emphasized on the nutritional value of plant milk. Male consumers presented a preference of plant milk with higher protein and enriched antioxidants, likely driven by their goals of muscle building and fitness. On the other hand, female consumers preferred plant milk with enriched collagen, dietary fiber, probiotics, and low-calorie, reflecting their focus on beauty and skincare. It can be seen that Chinese consumers are looking for products with higher nutritional value and lower calories, and willing to pay higher prices for beverages with improved quality and safety assurance. Due to the increasing incidences of obesity and diabetes in China, whole grains such as buckwheat, quinoa, and highland barley are promising raw materials for plant milk.

However, it is important to acknowledgethat a portion of consumers (34.9%) considered the current plant milk prices was expensive, making price a critical factor in their purchasing decisions. In contrast, there was a clear willingness among consumers to pay higher prices for plant milk products that offered enhanced nutritional value and product quality. To balance these two preferences and profit from the products at the same time, producers might consider providing two kinds of products: low-priced plant milk with basic nutrients and high-priced plant milk with higher content of nutrients or additional nutritional elements.

There are some limitations in the current study, such like the sample size was limited to 1052 participants and most of them were came from western south of China, which might not be fully representative of the entire Chinese population. Future research could consider a larger and more diverse sample to enhance the generalizability of the findings. Secondly, the study relied on self-reported data, which might be subject to response biases like social desirability bias. Employing additional data collection methods, such as interviews or focus groups, could provide more in-depth insights into consumer preferences and behaviors in the future studies.

## Figures and Tables

**Figure 1 foods-14-01240-f001:**
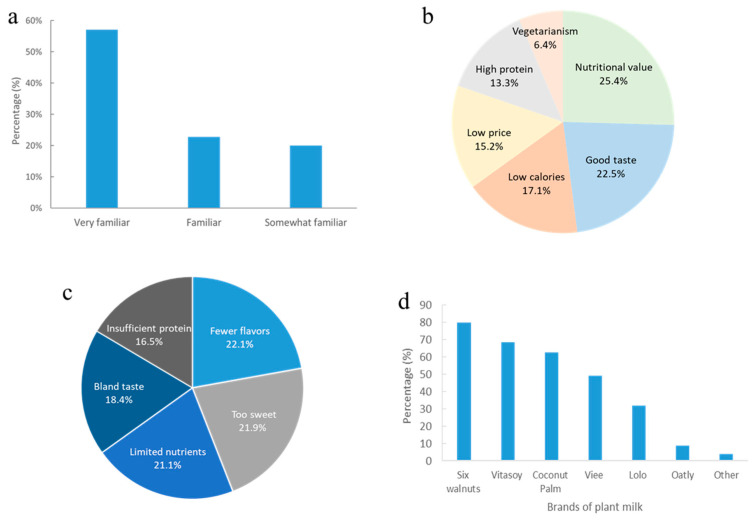
Cognitive analysis of consumers’ perception of plant milk including (**a**) familiarity with plant.milk, (**b**) the reasons for liking plant milk, (**c**) the defects of plant milk, and (**d**) awareness of different brands of plant milk.

**Figure 2 foods-14-01240-f002:**
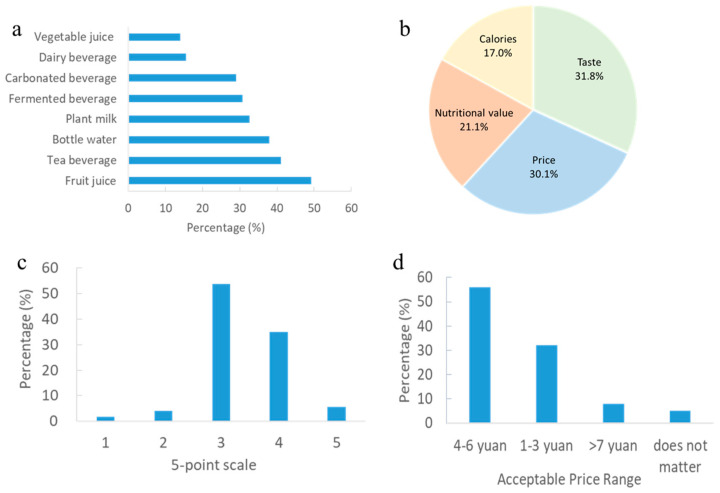
Analysis of purchase intention of consumers including (**a**) preference of beverage types, (**b**) decisive factors for beverage selection, (**c**) perception of the price of current plant milk rated by 5-point scale, and (**d**) perception of reasonable price of plant milk.

**Figure 3 foods-14-01240-f003:**
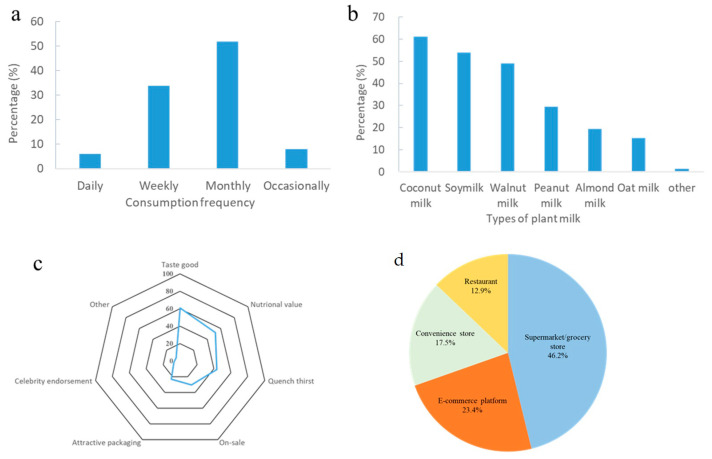
Analysis of purchase behaviors of consumers including (**a**) consumption frequency of plant milk, (**b**) types of plant milk of daily consumption, (**c**) decisive factors for selecting plant milk, and (**d**) location for purchasing plant milk.

**Figure 4 foods-14-01240-f004:**
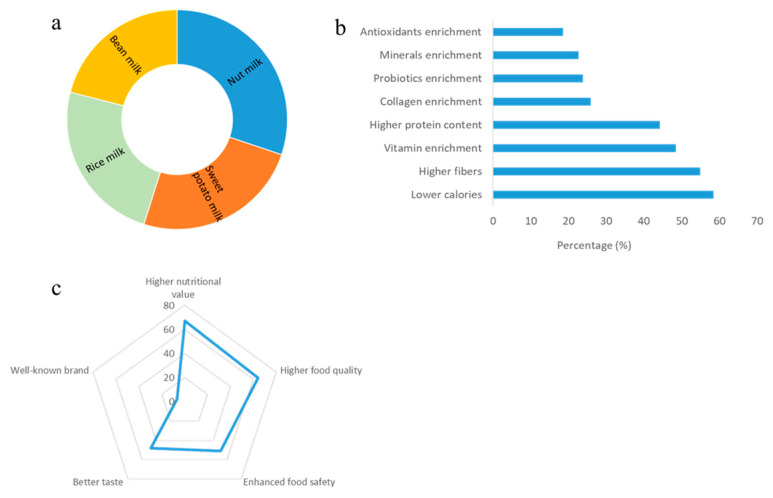
Analysis of future development and consumer’s preference of plant milk including (**a**) future demands of plant milk, (**b**) demand features of plant milk, (**c**) value-added attributes of plant milk.

**Figure 5 foods-14-01240-f005:**
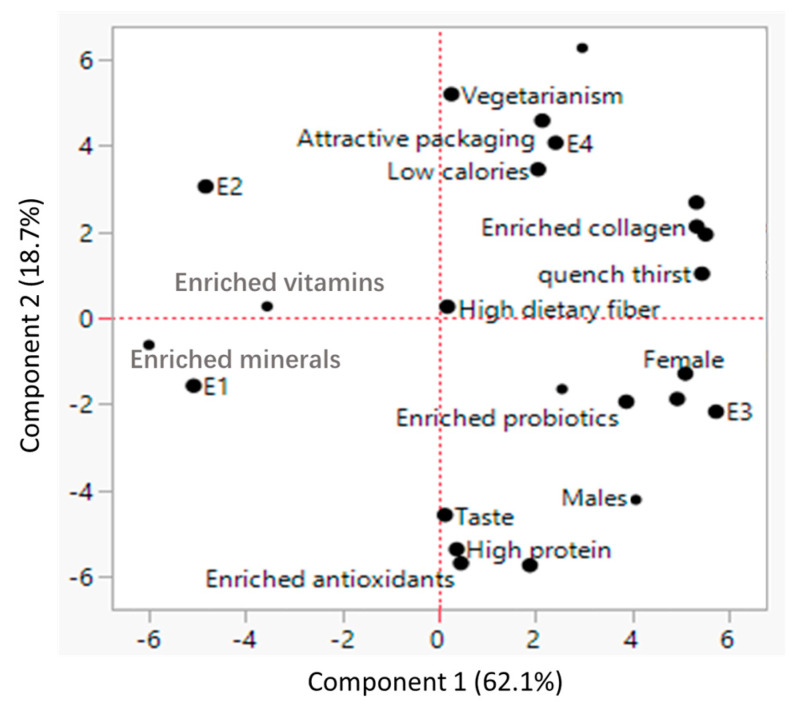
Principal component analysis of decisive factors for selecting plant milk (good taste, high protein, quench thirst, attractive packaging), demand features of plant milk (low calories, enriched antioxidants, enriched vitamins, enriched minerals, enriched probiotics, high dietary fiber, enriched collagen), and consumer population (gender, education level, vegetarianism). E1 means education level equal or below middle/high school, E2 means associate’s degree, E3 means bachelor’s degree, and E4 means master/doctor degree.

**Table 1 foods-14-01240-t001:** Descriptive statistics of the respondents (n = 1052).

Characteristics	Category	Frequency (n)	Percentage (%)
Gender	Male	406	38.6
	Female	646	61.4
Age	<18	74	7.0
	18–30	685	65.1
	31–40	175	16.6
	41–50	79	7.5
	>50	39	3.7
Education	Other	40	3.8
	Middle/High school	65	6.2
	Associate’s degree, occupational /academic	137	13.0
	Bachelor’s degree	705	67.0
	Master/Doctor	105	10.0
Occupation	Students	578	55.5
	Public servants	164	15.6
	Private enterprise employees	152	14.5
	Self-employed	63	6.6
	Farmers	29	2.8
	Retired	19	1.8
	Other	47	3.2
Monthly Salary(yuan)	<3000	620	58.9
	3000–6000	217	20.6
	6000–10,000	151	14.4
	>10,000	64	6.1
Location	Northern China	69	9.2
	Northeast China	52	6.9
	Northwest China	41	5.5
	Southwest China	311	41.4
	Southern China	80	10.6
	Central China	79	10.5
	Eastern China	120	16.0

**Table 2 foods-14-01240-t002:** Nutrition facts and price of major brands of plant milk in Chinese and American market ^1^.

Type	Brand	Content of Nutrients					Price (250 mL)
		Protein (g/100 mL)	Calorie (kcal/100 mL)	Fat (g/100 mL)	Carbohydrate (g/100 mL)	Vitamins (mg/100 mL)	
**China**							
Walnut milk	Six walnuts	0.6	40	2.6	3.7	N.A.	¥2.88
Soymilk	Vitasoy	2.0	47	1.4	6.5	1.27	¥2.09
Coconut milk	Coconut palm	0.6	51	2.3	7.0	N.A.	¥3.37
Peanut milk	Viee	1.0	38	1.8	4.4	0.7	¥4.08
Almond milk	Lolo	0.7	50	1.8	6.8	N.A.	¥3.56
Oat milk	Oatly	1.0	46	1.5	6.6	N.A.	¥5.20
**USA**							
Soymilk	Silk	3.3	46	1.9	3.8	65.0	$0.54
Coconut milk	Silk	0.0	29	2.1	2.5	77.5	$1.55
Almond milk	Silk	0.4	13	1.0	0.4	61.2	$0.41
Oat milk	Oatly	1.3	67	3.8	6.3	69.0	$0.69

^1^ The prices of plant milk were obtained from the largest Chinese e-commerce platforms (Taobao, https://www.taobao.com/, accessed on 30 November 2024) and American e-commerce platforms (Walmart, https://www.walmart.com/, accessed on 30 November 2024).

## Data Availability

The data presented in this study are available on request from the corresponding author due to the data supporting the reported results in this study involve human subjects and contain personal information. Due to privacy and ethical restrictions, the datasets cannot be made publicly available. Further details about the data are available from the corresponding author upon reasonable request and with appropriate institutional or ethical approval.

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
