# Peer review of "Consumer Preference and Purchase Intention for Plant Milk: A Survey of Chinese Market"

_foods, 2025, doi:10.3390/foods14071240_

Round 1
Reviewer 1 Report
Comments and Suggestions for Authors
Title of the manuscript: Consumer preference and purchase intention for plant milk: A survey of Chinese market
Although the topic of the manuscript is interesting, there are some issues which need to be addressed in the research before proceeding to the next stage.
First and foremost, I am not sure whether the topic of the manuscript aligns with the scope of this special issue considering that it focuses on consumer preferences for plant-based milk and not on recent advancements in processing technologies and functional properties of cereals and legumes.
Further, References must be numbered in order of appearance in the text. The Authors should follow the Instructions for Authors. Also, in a large part of the manuscript, the text is not justified.
The applied research method was a survey. How were the questions in the questionnaire constructed, did the authors use some model? Did the authors use a pilot survey to refine the questionnaire?
Ethical approval for the involvement of human subjects in the study was approved. How is the participation of minors in research regulated, given that they accounted for 7% of the sample?
In line 124 is stated that means and standard deviations are included in results presentation, but are not shown anywhere in the paper.
As it is stated in lines 149-151: ...” participants who were totally unfamiliar with plant milk had been excluded from this study. Figure 1a showed the participants’ familiarity of plant milk. The majority (79.9 %) of the respondents were familiar or somewhat familiar with the plant milk.”... does this mean that for further analysis, a sample smaller than 1052 participants was used, i.e., only 79.9% of the given number?
lines 145-147: ... “By asking targeted questions, the survey identified the factors that significantly impact consumers’ willingness to purchase plant-based milk.”... How were the factors defined when factor analysis, as a statistical method, was not used? How was the significance of impact determined?
It seems that the Discussion failed to discuss the research in depth and repeated the research Results without any critical discussion of previous related literature.
The literature cited is poor. It is suggested to include more literature and especially from prestigious scientific journals.
Please in Conclusion supplement your research limitations, and clarify the relationship between your limits and the future research.
The manuscript should be checked for typos and grammatical mistakes e.g. line 316: ... This variation might due to the...”
I wanted to recommend that the manuscript's English writing should be improved for clarity and flow. A polished version would help make the text more understandable and improve its overall readability.
Unfortunately, I consider that the experimental design, discussion and quality of presentation are not at the scientific level that characterizes the rank of this journal.
Comments on the Quality of English LanguageI wanted to recommend that the manuscript's English writing should be improved for clarity and flow. A polished version would help make the text more understandable and improve its overall readability
Author Response
Comments 1: First and foremost, I am not sure whether the topic of the manuscript aligns with the scope of this special issue considering that it focuses on consumer preferences for plant-based milk and not on recent advancements in processing technologies and functional properties of cereals and legumes.
Response 1:Thank you for your comments. While our study primarily focuse on consumer preferences and purchase intentions for plant-based milk, we believe it aligns with the special issue's scope in several ways. Our research identified key attributes such as nutritional value, taste, and calories, which directly relate to the functional properties of plant-based milk. Understanding these preferences provided the guidance for the development of new processing technologies to enhance these properties. Additionally, our findings highlight consumer interest in innovative plant milk products such as bean milk, rice milk, and quinoa milk. These insights can benefit the advancements in processing technologies to create novel plant-based milk products that meet diverse consumer needs. The study underscores consumers' willingness to pay for plant milk with enhanced nutritional value and improved product quality. This indicates a demand for advancements in processing technologies to fortify plant-based milk with additional nutrients and improve its overall quality. The information obtained from our study will drive targeted research on functional properties and processing technologies to cater to the specific demands of potential consumers.
Comments 2: Further, References must be numbered in order of appearance in the text. The Authors should follow the Instructions for Authors. Also, in a large part of the manuscript, the text is not justified.
Response 2: Thank you for your comments, we have revised the references as you suggested.
Comments 3: The applied research method was a survey. How were the questions in the questionnaire constructed, did the authors use some model? Did the authors use a pilot survey to refine the questionnaire?
Response 3: Thank you for your comments. The questions were designed to cover four main areas: cognitive analysis, purchase intention, purchase behavior, and user needs and product demands. To ensure the relevance and clarity of the questions, we conducted a thorough literature review and consulted with experts in the field of food science and consumer behavior. This helped us develop questions that accurately capture the factors influencing consumer preferences and behaviors regarding plant milk. Although we did not use a specific theoretical model to construct the questionnaire, our approach was guided by established research in the field. We have conducted a pilot survey with a small group of respondents (n=45), mainly the students of Food Science and Technology in Chengdu University with the age between 19-25 to refine the questionnaire. This pilot survey allowed us to identify and address any potential issues with question clarity, structure, and relevance. Based on the feedback from the pilot survey, we made necessary adjustments to improve the overall quality and reliability of the questionnaire.
Comments 4: Ethical approval for the involvement of human subjects in the study was approved. How is the participation of minors in research regulated, given that they accounted for 7% of the sample?
Response 4: Thank you for your comments. All participants, including minors, provided informed consent before taking part in the survey. Especially, for minors (participants under 18 years of age), we obtained informed consent from their parents. This process ensured that the minors' participation was voluntary and that their parents or guardians were fully aware of the study's purpose, procedures, and any potential risks. Additionally, we adhered to the guidelines set forth by the Chengdu University Research Ethics Committee to ensure that the rights and welfare of all participants, including minors, were protected throughout the study.
This study included minors because we once held a laboratory tour and study activity. We mentioned our survey research to the visiting parents and children. Parents and children who agreed to participate were informed and required to sign a consent form.
Comments 5: In line 124 is stated that means and standard deviations are included in results presentation, but are not shown anywhere in the paper.
Response 5: Thank you for your valuable feedback. We apologize for the oversight in our manuscript. The mention of means and standard deviations on line 124 was an error. Our study did not require the calculation of means and standard deviations for the data presented. We have corrected this in the manuscript to accurately reflect the statistical methods used in our analysis, which include descriptive statistics (frequency distribution tables, percentages) and principal component analysis (PCA). The revised manuscript ensures that all presented data aligns with the appropriate statistical methods employed.
Comments 6: As it is stated in lines 149-151: ...” participants who were totally unfamiliar with plant milk had been excluded from this study. Figure 1a showed the participants’ familiarity of plant milk. The majority (79.9 %) of the respondents were familiar or somewhat familiar with the plant milk.”... does this mean that for further analysis, a sample smaller than 1052 participants was used, i.e., only 79.9% of the given number?
Response 6: Thank you for your comments. We apologize for any confusion regarding the sample size used in our analysis. In our study, participants who were totally unfamiliar with plant milk were excluded from the survey before data collection. As a result, the actual valid sample size for our analysis is 1052 participants, all of whom were at least somewhat familiar with plant milk. Thus, in the Figure 1a, the number of respondent that was unfamiliar with plant milk was 0. We have revised the manuscript to explicitly state this point and ensure clarity.
Comments 7: lines 145-147: ... “By asking targeted questions, the survey identified the factors that significantly impact consumers’ willingness to purchase plant-based milk.”... How were the factors defined when factor analysis, as a statistical method, was not used? How was the significance of impact determined?
Response 7: Thank you for your comments. In our study, we identified the factors that significantly impact consumers' willingness to purchase plant-based milk through a combination of descriptive analysis and Principal Component Analysis (PCA). Although we did not use factor analysis as a statistical method, we defined the factors based on the results of descriptive statistics, which included frequency distribution tables and percentages. These statistics helped us identify the most frequently mentioned attributes that consumers considered important when making purchase decisions. The key factors identified were nutritional value, taste, calories, protein content, and price.
To determine the significance of the impact of these factors, we employed Principal Component Analysis (PCA), which allowed us to analyze the relationships and variances between multiple influencing factors. PCA helped us to reduce the dimensionality of the data and identify the principal components that explain the most variance in consumers' willingness to purchase plant-based milk.
Comments 8: It seems that the Discussion failed to discuss the research in depth and repeated the research Results without any critical discussion of previous related literature.
Response 8: Thank you for your comments. We have expanded the discussion section to further analyze the factors affecting preference, acceptance and types of plant-based milks and compared with other consumer studies, and incorporating additional insights on consumer preferences, sensory attributes, and the impact of social and environmental factors.
Comments 9: The literature cited is poor. It is suggested to include more literature and especially from prestigious scientific journals.
Response 9: Thank you for your comments. We have added new literatures in the manuscript.
Comments 10: Please in Conclusion supplement your research limitations, and clarify the relationship between your limits and the future research.
Response 10: Thank you for your comments. We have added the limitation and future researches in the Conclusion section of the manuscript.
Comments 11: The manuscript should be checked for typos and grammatical mistakes e.g. line 316: ... This variation might due to the...”
Response 11: Thank you for your comments. We have checked and revised the typos and grammatical mistake as you suggested.
Reviewer 2 Report
Comments and Suggestions for Authors
First of all Congratulations for your submission (I appreciate your work!) because the article contributes significantly to the understandings of the consumer preferences, purchase intentions, and behaviors related to plant milk in the Chinese market, focusing on factors like nutritional value, taste, and price. It also explores future consumer demands for innovative plant milk products, such as nut-based and low-calorie options.
The introduction is well written, but could benefit from more references to recent studies (e.g., from 2020 onwards) on consumer behavior and plant milk trends in other markets..
The research design seems appropriate to address the stated objectives and the use of an online questionnaire is suitable to collect data from a large and diverse sample of Chinese consumers, but although the survey covers a wide range of topics, it does not specify whether the questions have been pre-tested for clarity, reliability or validity.
The methods section of the paper provides a clear and structured description of the procedures followed but does not mention whether the survey questions have been validated (eg, by expert review or pilot testing) to ensure their reliability and relevance.
The results of the work are presented in a structured and logical manner, covering key aspects of the research objectives, but it can be observed throughout the work that the figures and tables, although informative, do not have a thorough interpretation in the text. For example, PCA analysis (Fig 5).
The conclusions in the paper are generally aligned with the results presented, but there are some gaps such as the thing that consumers are willing to pay a premium for high-quality products. However, the results also indicate that a significant portion of respondents view plant milk as expensive, and price was a decisive factor for many. You should address this "contradiction" .
Good luck
Comments on the Quality of English LanguageThe paper occasionally shifts between present and past tense inconsistently, particularly in the results and discussion sections. Please check this thing!
Author Response
Comments 1: First of all Congratulations for your submission (I appreciate your work!) because the article contributes significantly to the understandings of the consumer preferences, purchase intentions, and behaviors related to plant milk in the Chinese market, focusing on factors like nutritional value, taste, and price. It also explores future consumer demands for innovative plant milk products, such as nut-based and low-calorie options.
The introduction is well written, but could benefit from more references to recent studies (e.g., from 2020 onwards) on consumer behavior and plant milk trends in other markets.
Response 1: Thank you for your comments. We have revised the Introduction section as you suggested.
Comments 2: The research design seems appropriate to address the stated objectives and the use of an online questionnaire is suitable to collect data from a large and diverse sample of Chinese consumers, but although the survey covers a wide range of topics, it does not specify whether the questions have been pre-tested for clarity, reliability or validity.
Response 2: Thank you for your comments. We have conducted a pilot survey with a small group of respondents (n=45), mainly the students of Food Science and Technology in Chengdu University with the age between 19-25 to refine the questionnaire. This pilot survey allowed us to identify and address any potential issues with question clarity, structure, and relevance. Based on the feedback from the pilot survey, we made necessary adjustments to improve the overall quality and reliability of the questionnaire.
Comments 3: The methods section of the paper provides a clear and structured description of the procedures followed but does not mention whether the survey questions have been validated (eg, by expert review or pilot testing) to ensure their reliability and relevance.
Response 3: Thank you for your comments. To ensure the relevance and clarity of the questions, we conducted a thorough literature review and consulted with experts in the field of food science and consumer behavior. This helped us develop questions that accurately capture the factors influencing consumer preferences and behaviors regarding plant milk. Although we did not use a specific theoretical model to construct the questionnaire, our approach was guided by established research in the field. In addition, we have conducted a pilot survey to identify and address any potential issues with question clarity, structure, and relevance. Based on the feedback from the pilot survey, we made necessary adjustments to improve the overall quality and reliability of the questionnaire.
Comments 4: The results of the work are presented in a structured and logical manner, covering key aspects of the research objectives, but it can be observed throughout the work that the figures and tables, although informative, do not have a thorough interpretation in the text. For example, PCA analysis (Fig 5).
Response 4: Thank you for your comments. We have revised the Results and Discussion section as you suggested.
Comments 5: The conclusions in the paper are generally aligned with the results presented, but there are some gaps such as the thing that consumers are willing to pay a premium for high-quality products. However, the results also indicate that a significant portion of respondents view plant milk as expensive, and price was a decisive factor for many. You should address this "contradiction" .
Response 5: Thank you for your comments. To address this contradiction, we have revised the conclusion in the manuscript to better reflect the complexity of consumer attitudes towards plant milk pricing: “However, we cannot ignore the fact that a portion of consumers (34.9%) considered the current plant milk prices was expensive, making price a critical factor in their pur-chasing decisions. In contrast, there was a clear willingness among consumers to pay higher prices for plant milk products that offered enhanced nutritional value and product quality. To balance these two preferences and profit from the products at the same time, producers might consider providing two kinds of products: low-priced plant milk with basic nutrients and high-priced plant milk with higher content of nutrients or additional nutritional elements”.
Reviewer 3 Report
Comments and Suggestions for Authors
The article analyzes the preference for plant-based milks among a consumer sample of 1052 participants in China. However, there are some shortcomings in the manuscript that significantly weaken it.
What is the relevance and justification for conducting the study?
The literature review on factors affecting preference, acceptance and types of plant-based milks in consumer studies should be further explored.
The methodology should not only detail the questions used in the questionnaire, but also the justification of the questions based on the established literature. The questionnaire can be added as supplementary material. I am not sure that the PCA analysis is appropriate for the variables that were introduced in the model, as far as I can see, the variables are categorical.
In Figure 2, the relevance of the questions on beverage consumption behavior among the Chinese population is unclear, as no plant-based beverages milks are shown.
This relates to the following graphs which show that consumption of plant-based beverages milks is at a monthly frequency.
Overall, the study is very descriptive and should provide a more in-depth analysis of the variables and factors that influence the consumption of plant-based milks, as well as generate a discussion that will enrich the findings in light of other global research.
Some additional references that the authors may wish to consult are as follows:
https://doi.org/10.1111/ijfs.17517
https://doi.org/10.1016/j.foodres.2022.111648
https://doi.org/10.3168/jds.2016-12519
https://doi.org/10.1007/978-981-97-7870-6_11
https://doi.org/10.1016/j.foodqual.2022.104599
https://doi.org/10.1108/CAER-08-2023-0225
Comments on the Quality of English LanguageThe writing and editing of the manuscript should be revised.
Author Response
Comments 1: The article analyzes the preference for plant-based milks among a consumer sample of 1052 participants in China. However, there are some shortcomings in the manuscript that significantly weaken it.
What is the relevance and justification for conducting the study?
Response 1: Thank you for your comments. The relevance of our study lies in the rapid growth of the plant milk market in China and the increasing consumer interest in nutritious, sustainable, and vegetarian food options. Despite the significant market growth, there is a lack of research investigating the specific demands and purchase behaviors of Chinese consumers regarding plant milk. Understanding these consumer preferences is crucial for guiding the development of innovative and diversified plant milk products that can better meet the needs of the market. Especially after the 2008 melamine incident in China, the public lose the confidence in traditional dairy milk, leading to a shift towards plant-based alternatives. Our study aims to understand the factors influencing consumers' willingness to purchase plant milk, which can help manufacturers address consumer concerns related to health, safety, and nutrition.
Furthermore, plant milk has been the fastest-growing beverage category in China over the past decade, with a ten-year average growth rate of 24.5%. The market is expected to exceed 300 billion dollars by 2025. Our study provides valuable insights into consumer preferences and behaviors, which can guide the industry in developing products that align with market trends and consumer demands. In addition, the current plant-based beverages in the Chinese market are highly homogeneous, lacking innovation and diversification. Our study identifies consumer demands for new and innovative plant milk products, such as bean milk, rice milk, and quinoa milk. These insights can drive product development and diversification to satisfy the evolving needs of consumers. By addressing these factors, our study contributes to the broader understanding of consumer preferences in the plant milk market and provides actionable insights for industry stakeholders to enhance product development and market strategies.
Comments 2: The literature review on factors affecting preference, acceptance and types of plant-based milks in consumer studies should be further explored.
Response 2: Thank you for your comments. To address this, we have incorporated additional literature that provides a comprehensive understanding of consumer preferences and acceptance of plant-based milk alternatives. We have added this discussion into the manuscript.
Comments 3: The methodology should not only detail the questions used in the questionnaire, but also the justification of the questions based on the established literature. The questionnaire can be added as supplementary material. I am not sure that the PCA analysis is appropriate for the variables that were introduced in the model, as far as I can see, the variables are categorical.
Response 3: Thank you for your comments. We have revised the description of questionnaire as you suggested. We acknowledge that PCA is typically used for continuous variables and that its application to categorical variables requires careful consideration. However, other consumer studies have also used PCA on the correlation matrix of categorical variables. A similar approach was used in the study "Drivers of choice for fluid milk versus plant-based alternatives: What are consumer perceptions of fluid milk?" by McCarthy et al. (2017), where PCA was applied to analyze categorical variables related to consumer perceptions and preferences for fluid milk and plant-based alternatives. This method allows us to analyze the relationships and variances between multiple influencing factors.
Comments 4: In Figure 2, the relevance of the questions on beverage consumption behavior among the Chinese population is unclear, as no plant-based beverages milks are shown. This relates to the following graphs which show that consumption of plant-based beverages milks is at a monthly frequency.
Response 4: Thank you for your comments. We apologize for the confusion caused by the typo in Figure 2a. The term "protein beverage" should be "plant milk." Figure 2a is intended to show the market position and popularity of plant milk among all beverage types. This figure illustrates the relative popularity of different beverage types, including plant milk, and helps to contextualize the consumption behavior of Chinese consumers regarding plant milk.
Comments 5: Overall, the study is very descriptive and should provide a more in-depth analysis of the variables and factors that influence the consumption of plant-based milks, as well as generate a discussion that will enrich the findings in light of other global research.
Some additional references that the authors may wish to consult are as follows:
https://doi.org/10.1111/ijfs.17517
https://doi.org/10.1016/j.foodres.2022.111648
https://doi.org/10.3168/jds.2016-12519
https://doi.org/10.1007/978-981-97-7870-6_11
https://doi.org/10.1016/j.foodqual.2022.104599
https://doi.org/10.1108/CAER-08-2023-0225
Response 5: Thank you for your comments. We have incorporated additional literatures for a more in-depth discussion of our data and results as you suggested, and we have added this discussion into the manuscript.
Round 2
Reviewer 1 Report
Comments and Suggestions for Authors
In my comment 2 the following was stated: References must be numbered in order of appearance in the text. The Authors should follow the Instructions for Authors regarding References (reference numbers should be placed in square brackets [ ]). In the majority of the manuscript, this has not been done.
Author Response
Comment 1: In my comment 2 the following was stated: References must be numbered in order of appearance in the text. The Authors should follow the Instructions for Authors regarding References (reference numbers should be placed in square brackets [ ]). In the majority of the manuscript, this has not been done.
Response 1: Thank you for your comment. Actually, we have sorted the references in order of appearance in the text, but the software we used may not match the version of the Words, so the square brackets did not show in the test. We have now manually edited the references throughout the manuscript to ensure that the reference numbers are enclosed in square brackets, as required by the Instructions for Authors.
Reviewer 2 Report
Comments and Suggestions for Authors
Thank you for the responses!
Author Response
Comment: Thank you for your response.
Response: Thank you for reviewing the manuscript and providing the suggestion and guidance!
Reviewer 3 Report
Comments and Suggestions for Authors
The authors have addressed most of my comments.
The authors should describe in the methodology section that a correlation matrix was calculated on the category variables to perform the PCA.
Author Response
Comment 1: The authors should describe in the methodology section that a correlation matrix was calculated on the category variables to perform the PCA.
Response 1: Thank you for your comment! We have revised the description of methodology as you suggested. The updated methodology section now includes this clarification: “Prior to performing the PCA, a correlation matrix was calculated for the categorical variables to ensure appropriate data preparation”.